# Space-Time Evolution Analysis of the Nanjing Metro Network Based on a Complex Network

**Wei Yu [1]**  **, Jun Chen [2],\* and Xingchen Yan [1]**

1   College of Automobile and Traffic Engineering, Nanjing Forestry University, Longpan Road 159#, Nanjing 210037, China; yuweicar@163.com (W.Y.); xingchenyan.acad@gmail.com (X.Y.)
2   School of Transportation, Southeast University, Si Pai Lou 2#, Nanjing 210096, China
\*   Correspondence: chenjun@seu.edu.cn

**Abstract:** Many cities in China have opened a subway, which has become an important part of urban public transport. How the metro line forms the metro network, and then changes the urban traffic pattern, is a problem worthy of attention. From 2005 to 2018, 10 metro lines were opened in Nanjing, which provides important reference data for the study of the spatial and temporal evolution of the Metro network. In this study, using the complex network method, according to the opening sequence of 10 metro lines in Nanjing, space L and space P models are established, respectively. In view of the evolution of metro network parameters, four parameters—network density, network centrality, network clustering coefficient, and network average distance—are proposed for evaluation. In view of the spatial structure change of the metro network, this study combines the concept of node degree in a complex network, analyzes the starting point, terminal point, and intersection point of metro line, and puts forward the concepts of star structure and ring structure. The analysis of the space-time evolution of Nanjing metro network shows that with the gradual opening of metro lines, the metro network presents a more complex structure; the line connection tends to important nodes, and gradually outlines the city's commercial space pattern.

**Keywords:** complex network; metro; space-time; evolution

## 1. Introduction

The rapid development of China's economy has promoted the process of urban construction. Many cities have opened subways as an important means of improving the level of urban traffic and changing the urban spatial pattern. Urban metro networks can be abstracted as metro stations and lines, and on this basis, a complex network can be formed. How the urban metro network evolves into a part of the urban public transport network, and then affects the urban spatial pattern, is a problem worthy of study.

Table 1 shows a list of existing relevant research on metro network evolution. The research on the evolution of transportation network includes the establishment of models, the analysis of development stages, and the evolution process. Malinetskii et al. (2009) [1] proposed a dynamic model of traffic network evolution, whose dynamic behavior is the result of system self-organization. Ma et al. [2] (2015) tried to extend the theory of in-depth learning to the analysis of large-scale traffic network, and used the data from a taxi global positioning system to model and predict the evolution of traffic congestion. Chen et al. (2017) [3] discussed the long-term evolution characteristics of land transportation in the Beijing-Tianjin-Hebei area, and explained the five development stages of land transportation network by using a time scale of 100 years. Liu et al. (2018) [4] studied the influence of travel time selection on the daily dynamic evolution of traffic flow in traffic network, and studied the evolution process of network traffic flow by numerical experiments.

**Table 1.** A list of existing relevant research on metro network evolution.

| Number | Author | Main Work |
| --- | --- | --- |
| 1 | Malinetskii, G.G. | Dynamic model of traffic network evolution |
| 2 | Ma, X. | Modeling and predicting the evolution of traffic congestion |
| 3 | Chen, Y. | Discussing the long-term evolution characteristics of land transportation |
| 4 | Liu, S. | Studying the evolution process of network traffic flow by numerical experiments |
| 5 | Wandelt, S. | Analyzing the evolution of the national network of international air transport |
| 6 | Li, Z. | Studying the time evolution of European air transport system |
| 7 | Wu, X. | Analyzing the characteristics and dynamic changes of the spatial distribution of air transport utilization |
| 8 | Kim, J. | Analyzing the spatial evolution of Seoul Railway Station in Korea |
| 9 | Jiang, Y. | Proposing a bus maintenance strategy based on complex evolutionary method |
| 10 | Mohammed, A. | Describing a simulation model of bus network evolution based on GIS |
| 11 | Chen, X. | Proposing the average annual bus trip per capita and the average daily trip per bus of bus system |
| 12 | Huang, A. | Studying the intrinsic attributes and evolution mechanism of urban public transport network |
| 13 | Wu, J. | Proposing a dynamic evolution model of daily movement in urban railway network |
| 14 | Leng, B. | Analyzing the evolution of Beijing metro network in the process of development |
| 15 | Kim, H. | Discussing the four evolution stages of the Seoul metro system |

A typical network of urban external traffic includes the aviation network and the railway network. Wandelt et al. (2015) [5] analyzed the evolution of the national network of international air transport from 2002 to 2013 from the perspective of network physical topology and the traffic information functional network. Li et al. (2015) [6] studied the time evolution of European air transport system, analyzed the air navigation routing network and airport network, and found that the hub node existing in two network layers is the bottleneck of network development. Wu et al. (2018) [7] analyzed the characteristics and dynamic changes of the spatial distribution of air transport utilization in China's provinces and regions, and established a multiple linear regression model to forecast future airport passengers and cargo volume and the throughput growth rate of each province. Kim (2009) [8] analyzed the spatial evolution of Seoul Railway Station in Korea, and revealed the mutual adaptation between the architectural elements of Seoul Railway Station and the urban environment.

The most common traffic network within a city is the bus transport system network. Jiang et al. (2012) [9] proposed a bus maintenance strategy based on the complex evolutionary method, and established a mathematical model with the objective of minimizing the waiting time of passengers at the current station and the next station. Mohammed et al. (2013) [10] describe a simulation model of bus network evolution based on Geographic Information System (GIS), which successfully identifies bus lines prone to frequency changes. Chen et al. (2014) [11] proposed the average annual bus trip per capita and the average daily trip per bus, describing two aspects of bus system performance: travel intention and operation efficiency. Based on these two indicators, a variable coefficient model was used to analyze the impact of each variable on the travel intention and operational efficiency of different types of cities. Huang et al. (2016) [12] studied the intrinsic attributes and evolution mechanism of urban public transport network, and proposed a two-level evolution model to simulate the development of public transport lines from the perspective of complex networks.

Some studies involve the travel characteristics, growth patterns, and evolution stages of metro network evolution. Wu et al. (2013) [13] proposed a dynamic evolution model of daily movement, which can better reflect the characteristics of pedestrian travel in an urban railway network. Leng et al. (2014) [14] analyzed the evolution of the Beijing Metro Network in the process of development, proposed a new growth model consisting of expansion model and strengthening model, and defined and evaluated the network using a line-weighted network. Kim et al. (2015) [15] discussed the four evolution stages of the Seoul metro system from the vertical changes of network accessibility and reliability. With the rapid expansion of the network, accessibility and reliability increase with the passage of time, but with different speeds, spatial patterns are also different.

Many networks can be simplified into nodes and relationships among them, which constitute the basic elements of complex networks. The small-world effect of complex networks means that most of the nodes are very close to each other [16]. Public transport network is also a special complex network. Transportation stations or hubs constitute nodes, and transportation routing constitutes the relationship between nodes [17–20]. As a part of an urban public transport network, the study of

the metro network includes the modeling of urban metro network, and on this basis, the structural parameters and the relationship between the metro network and traffic are analyzed. There are also some studies on how to keep the subway network stable when attacked [21–24].

Metro construction and investment have a long-term nature, and sustainable development must be considered. Once the metro system is built, it is very difficult to renew and reform it. With the continuous development of society, the travel needs of residents may also change. The planning of a metro network needs to be carried out in stages and adjusted continuously. The existing research on metro networks focuses on the evaluation of network performance and robustness when attacked. When studying the evolution of metro networks, the related articles often discuss an abstract evolution stage, rather than evaluating it with specific parameters.

The Nanjing metro network mentioned in the paper is still in the stage of rapid development, and the metro data provided by Nanjing Metro Co., Ltd. are relatively clear and can be related to the actual development of urban land and analysis. On this basis, the evolution analysis of Nanjing metro network has practical reference value. Nanjing, as a developed city in China, takes subway planning as an important part of urban traffic network. From 2005 to 2018, a total of 10 metro lines were opened in Nanjing, and more and more metro lines are under construction.

Once the urban metro lines form a network, it can reflect the obvious characteristics of economies of scale. Large-scale metro networks that can provide more accessible services, enhance the attractiveness to passengers, and cope with passenger flow will increase passenger flow revenue. Through the corresponding traffic data, we can clearly see how the subway lines in Nanjing gradually formed the subway network and affected the spatial pattern of the city. In this paper, 10 metro lines of the Nanjing metro are taken as the research objects. The complex network model of the Nanjing metro is constructed by space L and space P methods, and four parameters are selected to analyze its performance changes. These four parameters can effectively quantify the evolution of Nanjing's metro network, from which we can see the changes of network size, density, center, and clustering.

The urban subway network also promotes the sustainable development of the city as a whole. Because of the improvement in transportation convenience brought by the metro network, the value of land around the metro station has risen, which promotes the development and construction of the surrounding areas. In this paper, the formation of a transportation hub is discussed by using the space L model combined with the Nanjing metro planning map. In view of the spatial structure of the Nanjing metro network, the concept of node degree is adopted in this study. The starting point, terminal point, and intersection point of metro lines are extracted to analyze the connection and change of the metro network to determine urban spatial patterns. In addition, star structure and ring structure are used to describe the main spatial structure of the urban metro network.

The analysis of space-time evolution of the Nanjing metro network shows that, with the gradual opening of Metro lines, the metro network presents a more complex structure, and the network parameters are constantly changing. Nanjing metro line connections tend to be important nodes to communicate with various economic regions. The metro network of Nanjing has gradually outlined the commercial spatial pattern of the city. The expansion of the metro network expands the scale of urban development. The metro network eases the pressure on passenger flow in the original central area of the city, and promotes the development of planned land around the city into a new commercial center. The service life of the metro is very long, so its appreciation potential is huge. This is also the reason why investors or residents choose to develop around the subway. This research provides an idea for studying the actual evolution of a metro network in an urban space.

## 2. Introduction to the Evolution of Nanjing Metro

### 2.1. Opening of Nanjing Metro

As of May 2018, the Nanjing metro network has opened 10 lines in the order of 1, 2, 10, S1, S8, 3, 4, S3, S9, and S7. The 10 lines are numbered 1-10, in this order shown in Figure 3 to Figure 10.

Table 2 shows the sequence and related performance of Nanjing metro. Line 1 opened in 2005, from Maigaoqiao to the Aotizhongxin, a total of 16 stations. Later, Line 1 opened the South Extension Line in 2010. These 27 stations were later collectively referred to as Line 1. S1 opened eight stations in 2014. In 2018, the S1 extension station ran from Nanjingnanzhan to Konggangxinchengjiangning. Line S7 was originally planned to be 10 stations, but later it was adjusted to nine stations when it was opened, and some stations have changed their names.

**Table 2.** Opening sequence and related performance of Nanjing metro.

| Opening Sequence | Number of Stations | Line Length (km) | Opening Year |
|---|---|---|---|
| 1 | 27 (16) | 38.9 | 2005 |
| 2 | 26 | 37.95 | 2010 |
| 10 | 14 | 21.6 | 2014 |
| S1 | 9 (8) | 37.3 | 2014 |
| S8 | 17 | 45.2 | 2014 |
| 3 | 29 | 44.9 | 2015 |
| 4 | 18 | 33.8 | 2017 |
| S3 | 19 | 36.22 | 2017 |
| S9 | 6 | 52.4 | 2017 |
| S7 | 9 (10) | 30.16 | 2018 |

*2.2. Telematics Evolution Exploited in the Nanjing Metro Network*

The operation of urban rail transit involves the interaction of a lot of information, so a multi-functional dispatching system with strong specificity, high reliability, and a complex interface is indispensable. In the initial stage of information construction, the Nanjing metro adopted a single system, focusing on the needs of various departments and centers, and solved practical problems with less investment, thus improving efficiency. However, with the transition from single-line operation to network operation, the problems of resource sharing and data interchange among different systems are becoming more and more serious, which seriously affects the efficiency and quality of work. There are many technical problems because the systems are developed by different software companies.

In 2005, when Nanjing metro Line 1 was opened, Terrestrial Trunked Radio (TETRA) wireless communication technology was adopted. With the continuous opening of Nanjing metro lines, the requirement for the wireless communication network of urban rail became higher and higher. Vulnerability to interference, difficult maintenance, and poor high-speed mobility are the three main problems. With the release of the LTE-M standard by the China Urban Rail Transit Association, Long Term Evolution (LTE) has become the mainstream direction of urban rail trains and ground wireless transportation. At the end of 2017, Nanjing metro Line S9 began to use the LTE vehicle-to-ground wireless system. LTE is used as the wireless transmission technology of urban rail transit vehicle-to-ground. It can ensure the high-speed and high-quality transmission of the following vehicle monitoring information and multimedia information, and provide a guarantee of improving the passenger environment of the Nanjing metro and improving the operational safety and efficiency.

The development of the Nanjing metro means higher requirements for a wireless communication system. If the wireless communication system cannot provide sufficient technical support for the operation of the Nanjing metro, the Nanjing metro network will not play its due role and its operational efficiency will not be improved. In the future, the train-to-ground wireless communication system of Nanjing metro can use the TETRA network for train dispatching and emergency dispatching. Meanwhile, a LTE network can be built to meet the needs of PIS and CCTV in the carriage. With the gradual application of the next generation of unmanned metro technology, reliability, comprehensive carrying capacity, and anti-interference ability of the vehicle-to-ground wireless technology are put forward as higher requirements. Only by the joint development of Nanjing metro network and telematics can a safer, green, and efficient urban rail transit system be built.

*2.3. Evolution of the Nanjing Metro Network and Sustainable Development of the City*

As an important part of the transportation system of big cities, the subway plays an active role in the sustainable development of cities. The planning of the Nanjing metro network needs to accurately grasp residents' traffic demands and coordinate with the overall development of the of the city to determine the nature, magnitude, and service level of the urban metro traffic demand, divide the functional levels, and select the appropriate rail transit mode.

With the gradual opening of Nanjing metro lines, the traffic system and business pattern of the city have changed accordingly. The important hub stations in the Nanjing metro network connect the metro with other modes of transportation such as buses, bicycles, and pedestrians, thus forming an integrated transportation system. Subway has become the preferred mode of public transport for Nanjing residents, which greatly relieves traffic pressure on the city.

When more metro lines were opened, the Nanjing metro gradually formed a complex network, connecting the city's commercial center with more remote areas. Residents began to choose to work in the city center during the day and take the subway at night to return to their relatively distant homes. For this reason, new residential and commercial centers began to emerge, usually along important metro lines. Nanjing metro has brought about the flow of people and commerce, which in turn has changed the status of urban land development. Flows of people in the urban center were dispersed to the suburbs to live, and airports and railway stations were relocated to the suburbs. The further improvement of the Nanjing metro network will greatly improve the living standards of residents.

## 3. Complex Network Modeling for Nanjing Metro Network

Data on the Nanjing metro line were released by the Nanjing Metro Group Co., Ltd. No consideration is given to the difference between the upstream and downstream lines of the metro. The station of the metro line shall be based on the actual operation. In building the model, the Nanjing metro network is assumed to be an undirected and unweighted network.

Both methods are based on Metro lines. The complex network of the Metro consists of nodes and edges, where the edges refer to the connections between nodes. The space L method defines a metro station as a node with a connection between two adjacent stations on any Metro line. If there is no other site between the two sites, they are connected to one side. The space L method reflects the geographic proximity of Metro stations. The space P method defines a metro station as a node with a connection between any two stations on any Metro line. If a subway line passes through the two stations, they are connected to one side. The space P method reflects the transfer relationship between Metro lines. These two modeling methods are to number the metro stations, and establish the relationship matrix between the stations through the metro lines [25]. On this basis, the performance parameters of the metro stations are analyzed by MATLAB R2012A software (MathWorks, Natick, America).

From 2005 to 2018, the sequence of Nanjing metro line opening is 1, 2, 10, s1, s8, 3, 4, s3, s9, s7. In this way, the complex network model of the Nanjing metro in different evolutionary stages is constructed, and the network parameters are analyzed and compared.

Through the relationship matrix between Nanjing metro sites, Netdraw 2 software (Analytic Technologies, Lexington, America) can draw Figures 1 and 2. Figure 1 shows the space L model of the complex network of Nanjing metro in 2018, which consists of 10 lines and 159 stations. From the model, it can be seen that the Nanjing metro network presents an overall structure of radiation from the center to the surrounding area. The ring structure in the center of the metro network outlines Nanjing's urban commercial pattern, with a star-shaped bifurcation structure around it, which communicates with airports, railway stations, and remote areas around it. The Nanjing metro network is still in the process of construction. In order to meet the needs of residents, more lines will be opened in the future, and the metro network will present a more complex structure.

Figure 2 shows the space P model of the complex network of the Nanjing metro. From the model, it can be seen that, because the model shows how all the stations on the same line are connected with each other, there are 10 clustering subgraphs in the graph, which actually represent 10 metro lines.

A connection between nodes on the same line is closer. These lines are connected by some important transit nodes.

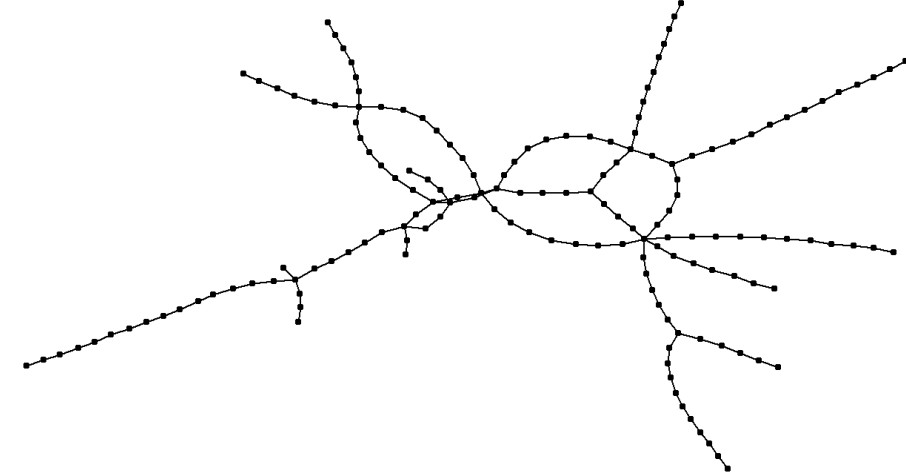

**Figure 1.** Space L model of Nanjing metro network.

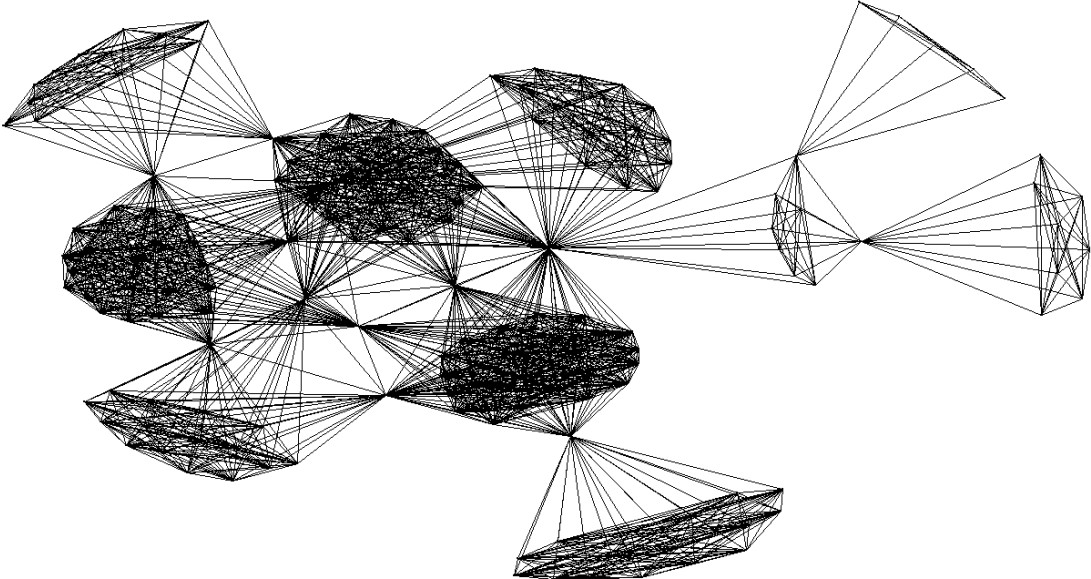

**Figure 2.** Space P model of Nanjing metro network.

## 4. Change of Performance Parameters of Complex Networks

In this study, four parameters—network density, network centrality, network average distance, and network clustering coefficient—are selected to evaluate the performance parameters of the Nanjing metro complex network. These four are typical parameters to measure complex traffic networks. In the complex network model of the Nanjing metro, the nodes represent the stations, and the connections between the nodes represent the subway lines passing through. Network density reflects the degree of close connection between stations in the metro network. The more metro lines there are between the same stations, the greater the network density. In a mature metro network, the connection between different metro stations is very close.

Network centrality reflects whether the nodes in the whole metro network are in the central position. Business centers and transportation hubs in stations often become the centers of Metro networks. This will also easily lead to network performance degradation when the central node is attacked. A multicenter model is a better solution to maintain the robustness of metro network. The network average distance reflects the actual distance or transfer times of different stations, and is

an effective indicator to reflect the level of passenger service. The smaller the average distance of the network, the more convenient it is for customers to travel on it. The network clustering coefficient reflects whether there is a small group effect in the metro network. In a mature metro network, as the links between stations become more intensive, some nodes form certain small groups, reflecting the situation of the geographical business circle or traffic circle.

*4.1. Parameter Selection*

(1) Network density

Network density refers to the degree of close connection between nodes in complex networks. The more network nodes are connected, the greater the network density. For undirected and unweighted networks, network density $d(G)$ is the ratio of the number of actual connections between the nodes in network $G$ to the maximum number of connections that may exist.

$$d(G) = \frac{2M}{[N(N-1)]} \tag{1}$$

In this formula, $M$ refers to the number of actual connections between nodes in a complex network. $N$ refers to the number of nodes. When the network is completely interconnected, the network density is 1.

(2) Network centrality

The centrality of complex networks reflects the relative importance of each node in the network. Node centrality of a network refers to the degree of centrality of a node in the neighbor nodes directly connected to it. Network centrality refers to the importance of a node or a group of nodes in the whole network, which shows the centralization of the whole network.

The degree of the node is related to the number of adjacent edges of the node. Node centrality $C_D(V_i)$ of node $V_i$ is the ratio of the actual degree to the maximum possible degree.

$$C_D(V_i) = \frac{k_i}{N-1} \tag{2}$$

In this formula, $k_i$ means the number of all adjacent nodes of node $i$, and $N$ means the number of all nodes.

In a network consisting of $N$ nodes, if there is a network $G_{optimal}$ that maximizes $H$:

$$H = \sum_{i=1}^{N} [C_D(V_{\max}) - C_D(V_i)]. \tag{3}$$

In this formula, $V_i$ means the node of the network $G_{optimal}$, and $V_{\max}$ means the node with the largest degree of centrality in network $G_{optimal}$.

For a network $G$ consisting of $N$ nodes, network centrality $C_D$ of network $G$ is defined as:

$$C_D = \frac{1}{N-2} \sum_{i=1}^{N} [C_D(V_{\max}) - C_D(V_i)]. \tag{4}$$

(3) Network average distance

The distance between two nodes in a network is called the distance between two nodes. Network average distance is the average distance between all nodes, which describes the degree of separation between nodes in the network.

For the undirected network, network average distance $L$ is defined as:

$$L = \frac{2}{N(N-1)} \sum_{i=1}^{N} \sum_{j=i+1}^{N} d_{ij}. \tag{5}$$

In this formula, $N$ is the total number of nodes, and $d_{ij}$ is the distance between node $i$ and node $j$.

(4) Network clustering coefficient

Network clustering coefficient refers to the average probability of interconnection between two nodes connected with the same node in the network, which can be used to represent the local structure of the network.

Network clustering coefficient $C$ is defined as:

$$C = \sum_{i=1}^{N} C_i = \sum_{i=1}^{N} \frac{2M_i}{k_i(k_i - 1)}. \tag{6}$$

In this formula, $C_i$ is the network clustering coefficient of node $i$, $N$ is the total number of nodes. For a undirected network, $k_i$ is the nodes connected directly with the node, and $M_i$ is the actual number of edges between the $k_i$ nodes.

The opening sequence of the Nanjing metro lines was 1, 2, 10, s1, s8, 3, 4, s3, S9, and s7. When space L and space P are analyzed, and the relationship between parameters and subway sequence is expressed by graphs, the 10 metro lines are set to 1 to 10 in sequence.

### 4.2. Variation of Performance Parameters of Space L Model

Figure 3 shows the evolution of space L network size in the Nanjing metro complex network. Network size refers to the number of stations in the subway network at all stages. The size of the space L of the metro network is the same as that of the space P. The size of space L is gradually increasing, and the size of space L is linearly related to the order of the subway. With the continuous opening of metro lines, more and more metro lines and stations have formed a complex Metro network.

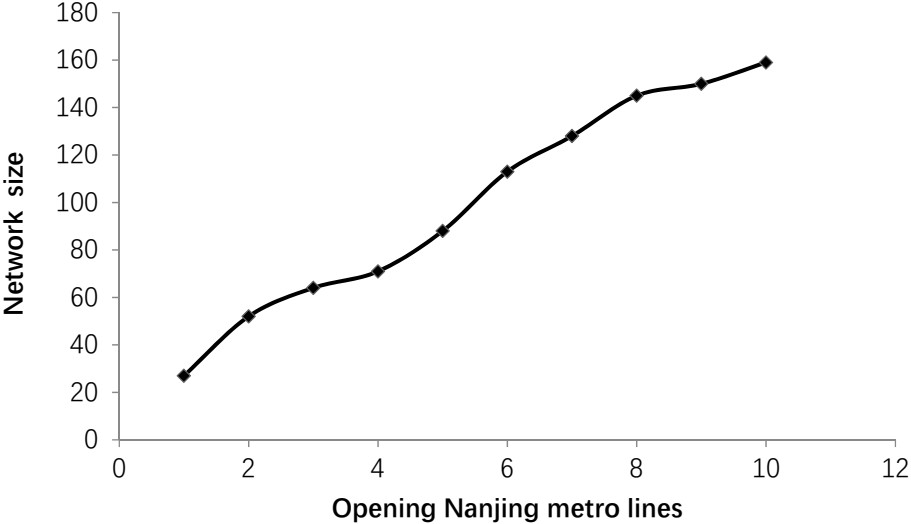

**Figure 3.** Evolution of space L network size in Nanjing metro complex network.

Figure 4 shows the evolution of the network density of space L in the complex network of the Nanjing metro. Network density means the degree of close connection between metro stations. The network density is between 0 and 1. The relationship between network density and the order of subway opening is the power law. With the increase in the number of metro lines and stations, the network density becomes smaller and smaller, and the connection between metro stations becomes looser. This is because the number of Nanjing metro lines is relatively small, and they are still spreading to the surrounding areas, forming fewer loops. Metro lines are also different from public transport lines, which play the role of the main line in the whole public transport network. As the number of metro stations tends to stabilize and the number of lines increases, the network density will begin to increase.

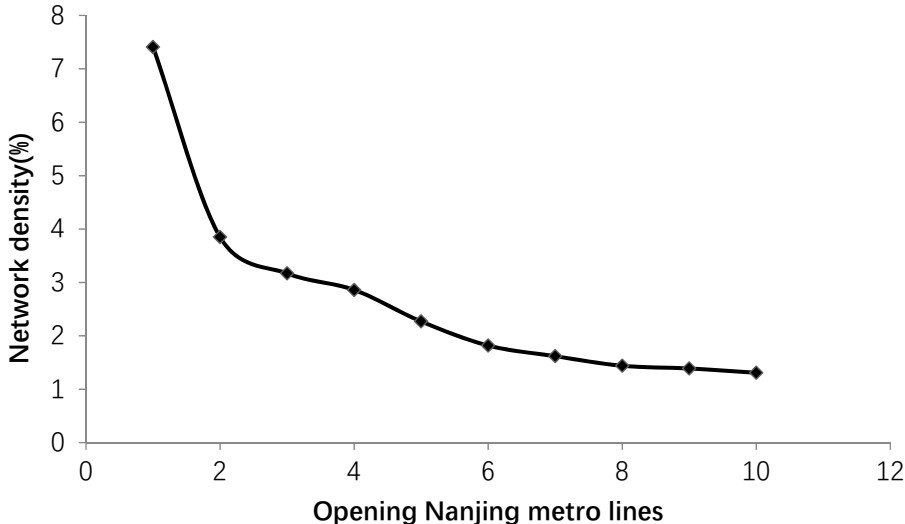

**Figure 4.** Evolution of the network density of Space L in Nanjing metro complex network.

Figure 5 shows the evolution of the network centrality of space L in the Nanjing metro complex network. For the space L of metro network, the connection between adjacent stations of the same line is established, which reflects the spatial relationship of stations. The network centrality of space L indicates whether there are obvious centers in the network, and geographically reflects the trend of convergence of metro lines to a certain center. The network centrality is between 0 and 1. As can be seen from Figure 5, when two metro lines are opened, the network centrality has an obvious peak and then begins to decline. Although there are some fluctuations, it tends to a stable value. This is because, after the intersection of Line 1 and Line 2 of Nanjing metro, a distinct center, Xinjiekou, was formed. Xinjiekou is not only the geographical center of Nanjing, but also the most important commercial center. With the opening of other metro lines, new centers began to emerge, dispersing the central effect of new streets, and the degree of network centrality began to decline. However, the subsequent opening of metro lines is still around the urban central area, so it will tend to a stable value.

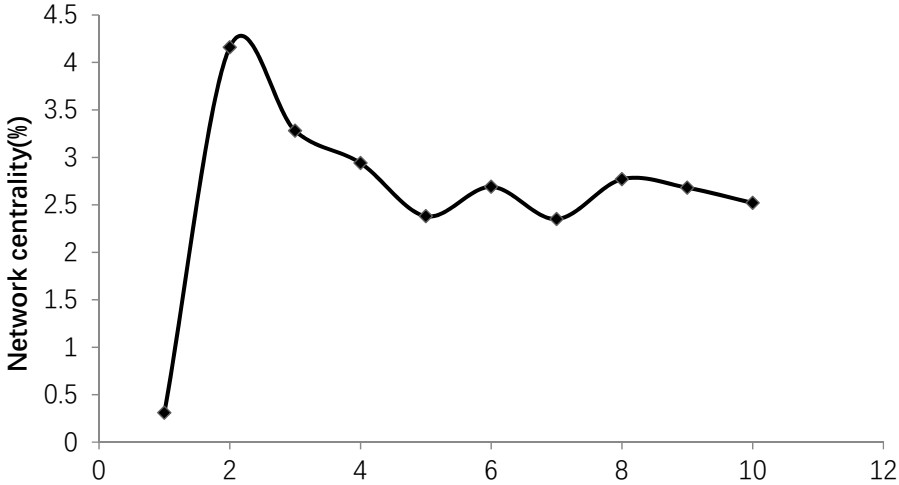

**Figure 5.** Evolution of the network centrality of space L in the Nanjing metro complex network.

Figure 6 shows the evolution of the average distance of the space L network in Nanjing metro complex network. For space L, network distance means the average number of sites between all sites. The average distance of the network and the number of Nanjing metro lines are linearly distributed. With the continuous increase of Nanjing metro lines, the average distance of the network is also increasing. This is because the Nanjing metro network is still in the stage of rapid expansion. With the

further increase in the number of metro lines, the existing metro lines will form a loop, thus reducing the network parity distance.

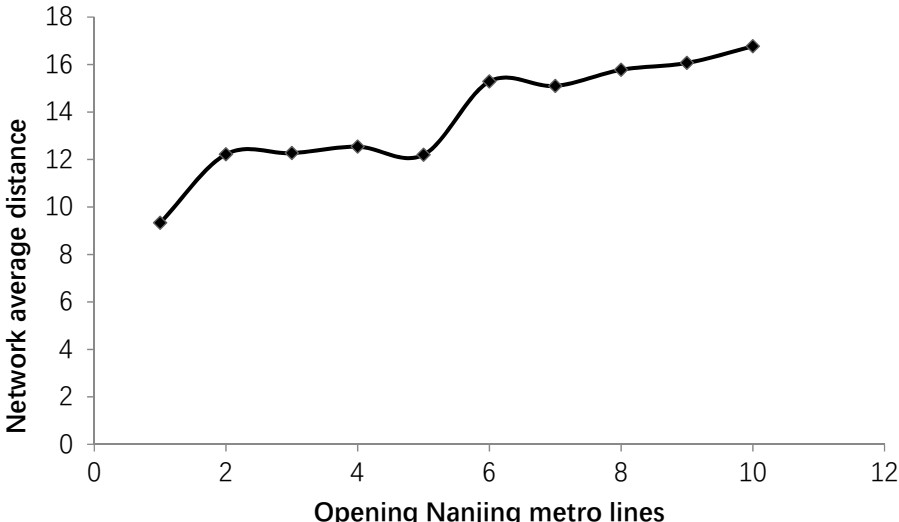

**Figure 6.** Evolution of the average distance of the space L network in Nanjing metro complex network.

For space L, the clustering coefficients of Nanjing metro network are close to 0. This is because the Nanjing metro network as a whole is still relatively loose, although there is a certain central point, but the central and surrounding areas have not yet formed a significant clustering effect.

*4.3. Variation of Performance Parameters of Space P Model*

Figure 7 shows the evolution of the network density of space P in Nanjing metro complex network. For space P, as long as a subway line passes between any two stations, the two stations will establish a connection, so the network density of space P is much higher than that of space L. The network density is between 0 and 1. When the first subway line is opened, there is a connection between any two stations, so the network density is 1 at the beginning, and then gradually decreases. The network density and the number of metro lines open present power law distribution. This is because, with the increase in Nanjing metro lines, there are links between stations in any line, but the links between lines become loose only through intersections.

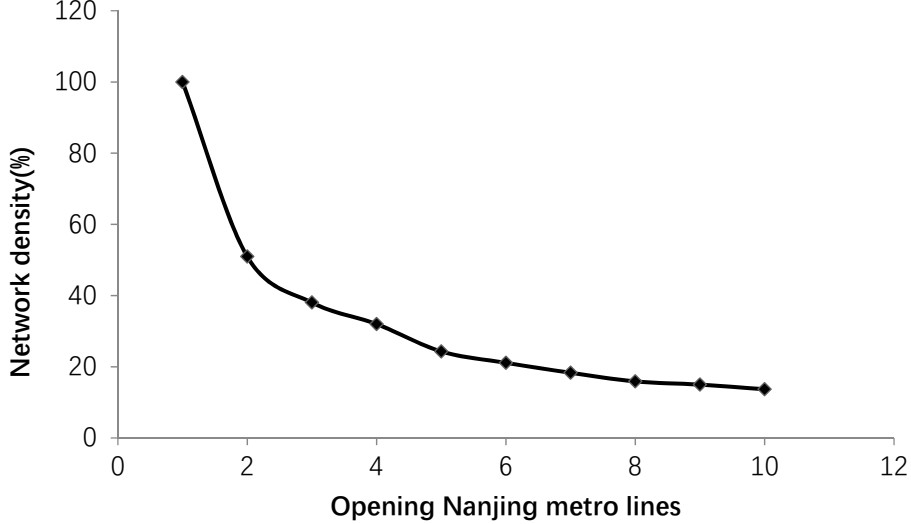

**Figure 7.** Evolution of the network density of space P in the Nanjing metro complex network.

Figure 8 shows the evolution of network centrality of space P in Nanjing metro complex network. Network centrality reflects whether the network is centralized to a certain center. The network centrality is between 0 and 1. When the first line is opened, because there is no center between any two stations, the network centrality starts at 0, reaches a peak when the second subway line is opened, then gradually decreases, and then slowly rises. This is because the center of the network represents the degree of intersection of lines. When two lines are opened, there is an obvious intersection, which is Xinjiekou. At this time, the network center degree is the highest. With the emergence of more intersections, the center of the network is decentralized. However, more other lines tend to connect to the original intersection, and the network centrality has improved.

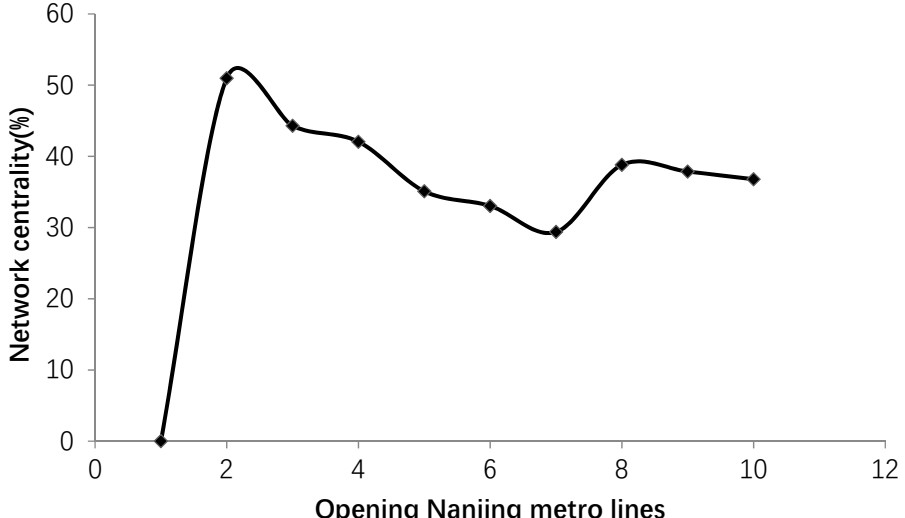

**Figure 8.** Evolution of network centrality of space P in the Nanjing metro complex network.

Figure 9 shows the evolution of network average distance of the space P network in the Nanjing metro complex network. For space P, network average distance means the number of average number of transfers of all stations. The network average distance of the network has a linear relationship with the order of subway line opening, which is gradually increasing. When the first line is opened, the number of transfers between any two stations is 1. With the increase in metro lines, the number of transfers between stations is also increasing, but the trend is relatively flat as a whole.

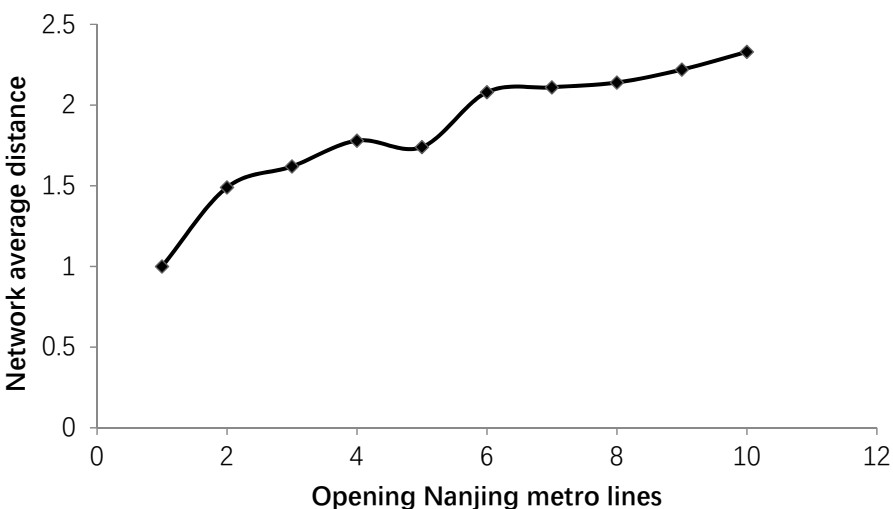

**Figure 9.** Evolution of network average distance of the space P network in the Nanjing metro complex network.

Figure 10 shows the evolution of the network clustering coefficient of space P in the Nanjing metro complex network. Network clustering coefficient indicates whether a certain clustering has been formed among stations. The network clustering coefficient is between 0 and 1. The network clustering coefficient has a linear relationship with the order of subway opening, which is gradually decreasing. When the first metro line is opened, any two stations are connected, so the network clustering coefficient is 1. With the increase in the number of metro lines, the metro network is more and more dispersed, and it is difficult to form clustering. However, when the number of metro stations reaches a stable value, the increased lines will combine some central nodes more closely, thus improving the network clustering coefficient.

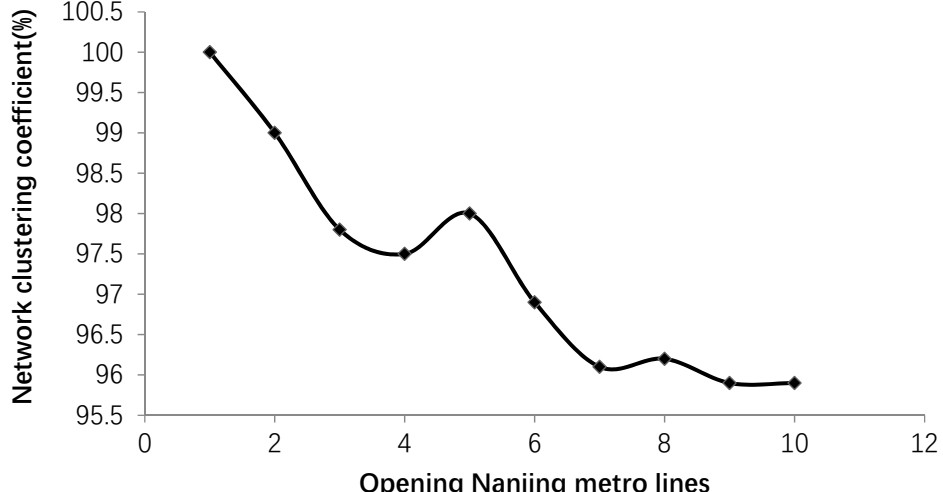

**Figure 10.** Evolution of the network clustering coefficient of space P in the Nanjing metro complex network.

## 5. Change of Spatial Topological Structure

This paper analyses the change of spatial topology structure of Nanjing metro network, mainly referring to the space L of complex network, which reflects the connection between adjacent nodes in geography. By comparing the topological structure map of space L with the spatial distribution map of the metro network, the changes of the spatial topological structure of Nanjing metro network can be seen by selecting important nodes and structures.

### 5.1. Important Nodes

Figure 11 shows the planning map of Nanjing metro line, which reflects 10 metro lines opened in Nanjing from 2005 to 2018. Different metro lines are represented by different colors, each has its own starting point and end point, and forms intersections with other metro lines, thus showing different topological structures. When Line S7 opened, Lukouxinchengdong changed name to Konggangxinchengjiangning.

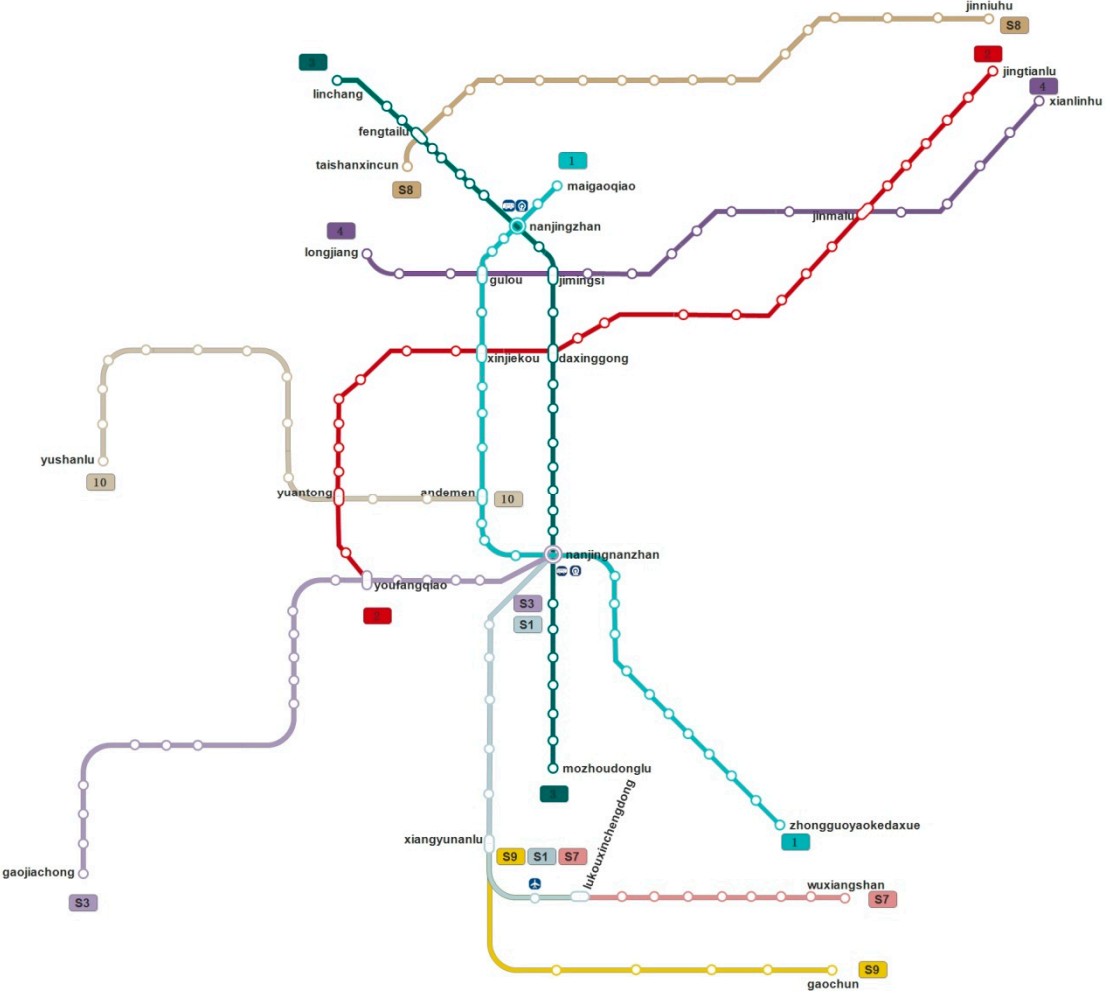

**Figure 11.** Planning map of Nanjing metro line.

Table 3 shows the important nodes in the Nanjing metro network, including the starting point, the end point, and the intersection point after the opening of different metro lines. When each new metro line is opened, the purpose of its planning is to try to connect remote areas or transport hubs and intersect with other existing metro lines to form a metro network. As of 2018, there are 11 districts in Nanjing, including Xuanwu, Qinhuai, Gulou, Jianye, Qixia, Yuhuatai, Pukou, Jiangning, Liuhe, and Lishui. As can be seen from Table 3, Nanjing metro lines connect these areas, centering on the main urban area and radiating to the surrounding areas.

The space L model structure of the Nanjing metro network shown in Figure 1 and the Nanjing metro line planning map shown in Figure 11 reflect the spatial connection and distribution of metro stations, and they have a good correspondence. Figure 1 shows the spatial model of the complex network of the Nanjing metro in 2018. In space L, a high degree of nodes indicate that more lines pass through. In the Nanjing metro line planning map shown in Figure 11, we can clearly see how the intersection points of the Metro line gradually formed.

Table 2 shows how intersections are formed when new metro lines are connected, thus connecting important transport hubs. These stations are important hubs in the metro network, connecting commercial centers, railway stations, airports, and remote suburban counties. When Lines 1 and 2 opened, a clear intersection was formed, Xinjiekou. Xinjiekou is located in the geographic center of the metro network and also the commercial center of the city. The two lines are cross-shaped, connecting the main areas of Nanjing.

**Table 3.** Changes of important nodes in Nanjing metro network.

| Opening Order | Starting Point | End Point | Connected Region | Intersection Lines | Intersection Point |
|---|---|---|---|---|---|
| 1 | Maigaoqiao | Zhongguoyaokedaxue | Qixia, Gulou, Xuanwu, Qinhuai, Yuhuatai, Jiangning | | |
| 2 | Youfangqiao | Jingtianlu | Jianye, Gulou, Qinhuai, Xuanwu, Qixia | 1 | Xinjiekou |
| 10 | Yushanlu | Andemen | Pukou, Jianye, Yuhuatai | 1, 2 | Andemen, Yuantong |
| S1 | Nanjingnanzhan | Konggangxinchengjiangning | Yuhuatai, Jiangning | 1 | Nanjingnanzhan |
| S8 | Taishanxincun | Jinniuhu | Pukou, Liuhe | | |
| 3 | Linchang | Mozhoudonglu | Pukou, Gulou, Xuanwu, Qinhuai, Yuhuatai, Jiangning | 1, 2, S1, S8 | Taifenglu, Nanjingzhan, Daxinggong, Nanjingnanzhan |
| 4 | Longjiang | Xianlinhu | Gulou, Xuanwu, Qixia, Jiangning | 1, 2, 3 | Jinmalu, Jimingsi, Gulou |
| S3 | Nanjingnanzhan | Gaojiachong | Jiangning, Yuhuatai, Jianye, Pukou | 1, 2, 3, S3 | Nanjingnanzhan, Youfangqiao |
| S9 | Xiangyulunan | Gaochun | Jiangning, Lishui, Gaochun | S1 | Xiangyulunan |
| S7 | Konggangxinchengjiangning | Wuxiangshan | Jiangning, Lishui | S1 | Konggangxinchengjiangning |

The opening of Line 10 has formed two intersections, Andmen and Yuantong, connecting the western region of Nanjing, which is also a new development center. The opening of Line S1 has formed the intersection of Nanjingnanzhan, which is a new railway station in Nanjing and an important passenger flow center. Line S1 also extended the metro line to Nanjing airport, the Konggangxinchengjiangning. Line S8 connects two remote areas of Nanjing, Pukou and Liuhe.

Line 3 has a longer line and more connecting areas, forming four intersections. Taifenglu station connects Line 3 with Line S8. Nanjingzhan station is an old railway station, which is connected with the new railway station, namely Nanjingnanzhan, through Line 3. Daxinggong is near Xinjiekou and belongs to the same core business circle. The opening of Daxinggong Station has dispersed the passenger flow pressure of Xinjiekou Station.

Line 4 intersects with three main lines, namely Line 1, Line 2 and Line 3, which further improves the spatial layout of the Nanjing metro network. At the intersection of Line 4, Gulou and Jimingsi are near the new street junction, and the core circle is further linked by the metro.

Line S3, starting from Nanjingnanzhan, further links the area with the new railway station. Lines S7 and S9 connect several remote areas in Nanjing. These remote areas are more convenient for new railway stations and airports, which paves the way for the future commercial development of land.

*5.2. Important Structure*

With the continuous opening of metro lines, the structure of the metro network is becoming more and more complex. The interconnection of metro lines forms some important intersections and different network structures. These structures are mainly star structure and ring structure.

Star structure is a structure that radiates outward from a central point. Generally, there are two kinds, one is the star structure with the transportation hub as the core, the other is the star structure at the beginning or end of the line, which communicates with relatively remote areas. Nanjing's new railway station, Nanjingnanzhan, and its surrounding lines constitute a typical complex star

structure. Nanjingnanzhan station plays an important transit role between urban and suburban areas. Other nodes such as Taifenglu, Andermen, Yuantong, Youfangqiao, Xiangyulunan, and Xianhemen also have a simple star structure, which communicates with the airport and some remote suburbs.

The ring structure is composed of some closed rings through metro lines. A ring structure is generally in the center of the city. These ring structures often coincide with the outline of the city's commercial areas. With the continuous increase of metro lines, the ring structure in the metro network is more and more common, and the centrality is more obvious. For example, the main lines of the Nanjing metro are Lines 1-4. After the four lines intersect with each other, they form four intersections and a ring structure. The four stations of Xinjiekou, Gulou, Daxinggong, and Jiminsi are located on the four corners of the ring structure, which delineates the core commercial area of Nanjing.

These four intersections, together with the two intersections of Nanjingzhan and Nanjingnanzhan, delineate the structure of the main urban area of Nanjing. The ring structure of the main urban area, together with the two intersections of Youfangqiao and Jinmalud, includes the two new development areas of Aoti and Xianlin. This larger ring structure reflects the outline of the urban area of Nanjing. With the opening of more metro lines, the ring structure will become more and more common, and the urban landform of Nanjing will be more delicately divided.

## 6. Conclusions

Many cities in China have opened subways, which have become an important part of urban public transport. How the metro line forms the metro network, and then changes the urban traffic pattern, is a problem worthy of attention. With the continuous opening of metro lines, a complex urban metro network has been formed. The urban transportation system has become more perfect, and the urban land development pattern has also changed. From 2005 to 2018, a total of 10 metro lines were opened in Nanjing, and more lines are being planned. The formation and development of the Nanjing metro network can be used as a reference sample to study the space-time evolution of the metro network. In this study, two models of space L and space P of Nanjing metro network are constructed. The size of the two networks is increasing gradually, and the order of opening subway lines shows a linear relationship.

For space L, with the continuous opening of metro lines, the network density is getting smaller and smaller, and the connection between metro stations becomes looser. The degree of network centrality first reaches a peak value, and then begins to decline gradually, indicating the transition from one center to multiple centers. The average distance of the network is also increasing, because the Nanjing metro is still in the stage of rapid development and tends to connect the surrounding remote areas. The clustering coefficients of the network are close to 0, and there is no obvious clustering effect in the Metro network.

For space P, the network density decreases gradually. There are fewer intersections between lines, and there are not too many loops. Network centrality first reaches a peak, then decreases and then slowly recovers. This is because there was an obvious center, but later it became multiple centers. When new lines opened, they tended to connect new centers. The average distance between the network and the metro line is gradually increasing. With the increase in metro lines, the number of transfers between stations is also increasing. The network clustering coefficient is gradually decreasing, and the nodes in the metro network are more and more dispersed. However, when the number of metro stations reaches a stable value, the increased lines will combine some central nodes more closely, thus improving the network clustering coefficient.

This paper analyses the change in topology of the Nanjing metro network, mainly referring to the space L model of complex network, and describes it with important nodes and important structures. In this model, a high degree of nodes indicates that more lines are passing through. In the Nanjing metro line planning map shown in Figure 11, we can clearly see how the intersection points of the metro line gradually formed. Stations as important nodes are also important hubs in the metro network, connecting commercial centers, railway stations, airports, and remote suburban counties. The main

structures in the Nanjing metro network are star structure and ring structure. The ring structure forms some closed rings through metro lines, which depicts the outline of urban areas. With the continuous increase in metro lines, the ring structure in the metro network is becoming more and more common, and the centrality is more obvious.

Previous studies of the evolution of metro networks often lack quantitative indicators, and it is difficult to compare the different stages of the evolution of metro networks. In this paper, four network performance parameters are used to analyze the evolution of the Nanjing metro network, which provides a quantitative analysis method for the evolution of the metro network. This method can also be applied to other traffic network analysis.

In the past, there was a lack of graph theory research on the evolution of the structure of the metro network. This paper establishes the space P model of the complex network, compares it with metro planning, and puts forward two typical network structures, star and ring. This can effectively describe the evolution of a metro network in actual geographical space, and provide effective ideas for the comprehensive development of transportation.

By May 2018, Nanjing had opened 10 routes. These lines constitute the complex network of the Nanjing metro, which changed the traffic pattern of the city and brought about changes in the planning pattern of Nanjing. In the future, more Nanjing metro lines will be planned and constructed, the Nanjing metro network will become more complex, and the spatial pattern of the Nanjing metro network will change accordingly. All of these need to be further studied in the future.

**Author Contributions:** J.C. undertook the data collection. W.Y. provided an interpretation of the results and wrote the majority of the paper. X.Y. contributed to the paper's review and editing.

**Funding:** This research was funded by a Key Project of the National Natural Science Foundation of China (grant no. 51638004), and the Basic Research Program of Science and Technology Commission Foundation of Jiangsu Province (grant no. BK20180775).

**Acknowledgments:** The authors would like to express their sincere thanks to the anonymous reviewers for their constructive comments on an earlier version of this manuscript.

**Conflicts of Interest:** The authors declare no conflict of interest.

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
