# Peer review of "Space‒Time Evolution Analysis of the Nanjing Metro Network Based on a Complex Network"

_sustainability, doi:10.3390/su11020523_

Round 1
Reviewer 1 Report
This article does not show a strong scientific relevance because it does not highlight an innovative method for evaluating data and scenarios. Figure 2 does not understand from which simulation/schematization is obtained Table 2 should be better introduced in the paragraph highlighting the information obtained The conclusions, as well as the discussion of the results, are lacking
Author Response
Point 1: This article does not show a strong scientific relevance because it does not highlight an innovative method for evaluating data and scenarios.
Response 1: The paper has been revised to supplement the description of data sources and scenarios.
Point 2: Figure 2 does not understand from which simulation/schematization is obtained.
Response 2: The models of space L and space P are supplemented in this paper. These two modeling methods are to number the metro stations, and establish the relationship matrix between the stations through the metro lines. On this basis, the performance parameters of the metro stations are analyzed by MATLAB software. Through the relationship matrix between Nanjing metro sites, Netdraw software can draw Figures 1 and 2.
Point 3: Table 2 should be better introduced in the paragraph highlighting the information obtained.
Response 3: The data in Table 2 and its meaning are supplemented in this paper.
Point 4: The conclusions, as well as the discussion of the results, are lacking.
Response 4: The conclusion and discussion are supplemented in this paper.

Reviewer 2 Report
The presented research is interesting. However, I have the following
questions.
What is the new contribution to science?
Are similar studies known in the literature?
And if so, what are the differences between those presented in the article?
What is the scientific purpose of the article and what are the scientific conclusions from the presented results?
Methodology description is missing, especially the methodology concerning Space L method and Space P method.
- lines 169-198: there are no explanations to the formulas, e.g. "Goptimal", "G", and so on.
- lines 169-198: the formatting of variables in the text should be arranged according to the formatting of the formulas. E.g: formatting the text "V max" does not match the formatting of the formula (3) and so on.
- lines 169-198: the explanations are not directly on the formulas but much further, which makes it difficult to interpret and understand the content. For example, the designation "ki" for the formula "(2)" (line 171) is too far because in line 197 and additionally on the next page.
- Figures 3-10 : There are no units on the axes. What does the term "Network size" mean? What does the term "Network density" mean? And so on. The axis description should be extended.
Author Response
Point 1: What is the new contribution to science?
Response 1: In this paper, ten metro lines in Nanjing metro are taken as the research objects. The complex network model of Nanjing metro is constructed by space L and space P methods, and four parameters are selected to analyze its performance changes. These four parameters can effectively quantify the evolution of Nanjing metro network, from which we can see the changes of network size, density, center and clustering.
In this paper, the formation of transportation hub is discussed by using space L model combined with Nanjing metro planning map. In addition, two typical spatial structures in Metro network, ring structure and star structure, are proposed to evaluate the overlapping effect of network and urban business circle. This provides a research idea for studying the actual evolution of metro network in urban geographic space.
The paper gives a supplementary explanation to this point.
Point 2: Are similar studies known in the literature?
Response 2: This paper summarizes the literature on the modeling and evolution of metro network.
Point 3: And if so, what are the differences between those presented in the article?
Response 3: The existing research on metro network focuses on the evaluation of network performance and robustness when attacked. When studying the evolution of metro network, the related articles often discuss an abstract evolution stage, rather than evaluating it with specific parameters. The Nanjing metro network mentioned in the paper is still in the stage of rapid development, and the metro data provided by Nanjing Metro Co., Ltd. are relatively clear, which can be related to the actual development of urban land and analysis. On this basis, the evolution analysis of Nanjing metro network has practical reference value.
The paper gives a supplementary explanation to this point.
Point 4: What is the scientific purpose of the article and what are the scientific conclusions from the presented results?
Response 4: This paper uses four network performance parameters to analyze the evolution of Nanjing metro network, and provides a quantitative analysis method for the evolution of the metro network. Combining the spatial structure model with the metro planning map, the two spatial structures proposed in this paper can effectively depict the evolution of metro network in the actual geographic space, and provide an effective idea for the comprehensive development of land and transportation.
The paper gives a supplementary explanation to this point.
Point 5: Methodology description is missing, especially the methodology concerning Space L method and Space P method.
Response 5: The construction method and explanation of this model have been supplemented in this paper.
Point 6:
- lines 169-198: there are no explanations to the formulas, e.g. "Goptimal", "G", and so on.
- lines 169-198: the formatting of variables in the text should be arranged according to the formatting of the formulas. E.g: formatting the text "V max" does not match the formatting of the formula (3) and so on.
- lines 169-198: the explanations are not directly on the formulas but much further, which makes it difficult to interpret and understand the content. For example, the designation "ki" for the formula "(2)" (line 171) is too far because in line 197 and additionally on the next page.
Response 6: The format of the above chart and formula has been adjusted and the parameters have been supplemented.
Point 7: Figures 3-10 : There are no units on the axes. What does the term "Network size" mean? What does the term "Network density" mean? And so on. The axis description should be extended.
Response 7: The parameters in the figures have been supplemented in this paper.

Round 2
Reviewer 1 Report
The revisions applied have almost completely concluded the information that was previously lacking in your work.
I think it is necessary to strengthen the motivations of your work and to better motivate the choice of the method of analysis used
We need to insert a greater motivation for the choice of the parameters that you have compared and described
In final paragraph (conclusion), I do not know anything scientifically advanced or innovative but only preliminary.
I also think the correlation through histograms is better in representing the different 4 parameters analyzed for the various transport lines
Author Response
Point 1: We need to insert a greater motivation for the choice of the parameters that you have compared and described.
Response 1: These four parameters are typical parameters to measure complex traffic networks. In the complex network model of Nanjing metro, the nodes represent the stations, and the connections between the nodes represent the subway lines passing by. Network density reflects the degree of close connection between stations in the metro network. The more metro lines between the same number of stations, the greater the network density. In a mature metro network, the connection between different metro stations is very close.
Network centrality reflects whether the nodes in the whole metro network are in the central position. Business centers and transportation hubs in stations often become the centers of Metro networks. This will also easily lead to network performance degradation when the central node is attacked. Multicenter model is a better solution to maintain the robustness of metro network. The network average distance reflects the actual distance or transfer times of different stations, and it is an effective indicator to reflect the level of passenger service. The smaller the average distance of the network, the more convenient it is for customers to travel. The network clustering coefficient reflects whether there is a small group effect in the metro network. In a mature metro network, as the links between stations become more intensive, some nodes form certain small groups, reflecting the situation of the geographical business circle or traffic circle.
The above explanation has been added to the paper.
Point 2: In final paragraph (conclusion), I do not know anything scientifically advanced or innovative but only preliminary.
Response 2: Previous studies on the evolution of metro networks often lack quantitative indicators, and it is difficult to compare the different stages of the evolution of metro networks.In this paper, four network performance parameters are used to analyze the evolution of Nanjing metro network, which provides a quantitative analysis method for the evolution of the metro network. This method can also be applied to other traffic network analysis.
In the past, there was a lack of graph theory research on the evolution of the structure of the metro network. This paper establishes the space P model of complex network, compares it with metro planning, and puts forward two typical network structures, star and ring. This can effectively describe the evolution of metro network in the actual geographical space, and provide effective ideas for the comprehensive development of land transportation.
The above explanation has been supplemented in the conclusion.
Point 3: I also think the correlation through histograms is better in representing the different 4 parameters analyzed for the various transport lines.
Response 3: In my opinion, the trend of these four parameters of Nanjing metro network is more appropriately represented by a curve graph, which can show the effect of direct fluctuation and some functional relationships.

Reviewer 2 Report
The article presents an interesting issue of metro network assessment. The authors introduced additional explanations and corrections to the text, which enriched the substantive quality. The explanations and corrections are satisfactory.
Author Response
Point 1: The article presents an interesting issue of metro network assessment. The authors introduced additional explanations and corrections to the text, which enriched the substantive quality. The explanations and corrections are satisfactory.
Response 1: According to the suggestions of other reviewers, some minor adjustments and explanations have been made to the paper.
